# Emergence of Weyl fermions by ferrimagnetism in a noncentrosymmetric magnetic Weyl semimetal

Cong Li [1,2,8] ✉, Jianfeng Zhang [3,8], Yang Wang[1,8], Hongxiong Liu[3,8], Qinda Guo [1], Emile Rienks [4], Wanyu Chen[1], Francois Bertran [5], Huancheng Yang[6], Dibya Phuyal[1], Hanna Fedderwitz[7], Balasubramanian Thiagarajan[7], Maciej Dendzik [1], Magnus H. Berntsen [1], Youguo Shi[3], Tao Xiang [3] & Oscar Tjernberg [1] ✉

Condensed matter physics has often provided a platform for investigating the interplay between particles and fields in cases that have not been observed in high-energy physics. Here, using angle-resolved photoemission spectroscopy, we provide an example of this by visualizing the electronic structure of a noncentrosymmetric magnetic Weyl semimetal candidate NdAlSi in both the paramagnetic and ferrimagnetic states. We observe surface Fermi arcs and bulk Weyl fermion dispersion as well as the emergence of new Weyl fermions in the ferrimagnetic state. Our results establish NdAlSi as a magnetic Weyl semimetal and provide an experimental observation of ferrimagnetic regulation of Weyl fermions in condensed matter.

In recent decades, considerable efforts have been made to identify topological quasiparticles in condensed matter physics that follow the same physical laws as elementary particles[1–7]. The discovery of topological semimetals has made this possible[8–20]. Weyl semimetals are an important class of topological semimetals, which host emergent relativistic Weyl fermions in the bulk, and Fermi arc surface states connect two Weyl points with opposite chirality on the boundary of a bulk sample[1,6,7,13–15]. These emergent particles are the result of inversion symmetry (IS)[6,7,13–15,21,22] or time-reversal symmetry (TRS)[1,2,23–32] breaking. The TRS-breaking Weyl semimetals are usually derived from magnetic materials[1,2,23–32], compared to the Weyl semimetals with IS-breaking, which provide a platform for the study of the interplay between magnetism, electron correlation and topological orders. In the magnetic Weyl semimetals, a nonvanishing Berry curvature induced by TRS-breaking can give rise to rich phenomena, such as an anomalous[33–35] and spin[36,37] Hall effect, and chiral anomalous

charge[17–20]. These exotic physical properties make magnetic Weyl semimetals potential candidates for a wide range of applications in spintronics.

In general, the establishment of Weyl semimetal must break either inversion or time-reversal symmetry. Recently, a rare case of magnetic Weyl semimetal candidates, the RAlX (R: Rare earth; X: Si or Ge) family, has attracted attention. This family displays both IS and TRS breaking[38–63]. From the perspective of crystal structure, RAlX is a non-centrosymmetric crystal with magnetism derived from R (Rare earth) atoms[38]. The magnetic structure of the RAlX family is easy to regulate and presents diverse magnetic ordering. For example, the magnetic structure of RAlX can be nonmagnetic[64–66], ferromagnetic[38–49], antiferromagnetic[49–56], ferrimagnetic[57–60] and even spiral magnetic[57,61] by rare earth element substitution. Above the magnetic transition temperature, the RAlX family are already an IS broken Weyl semimetal. When the temperature is lowered below

[1]Department of Applied Physics, KTH Royal Institute of Technology, Stockholm 11419, Sweden. [2]Department of Applied Physics, Stanford University, Stanford, CA 94305, USA. [3]Beijing National Laboratory for Condensed Matter Physics, Institute of Physics, Chinese Academy of Sciences, Beijing 100190, China. [4]Helmholtz-Zentrum Berlin für Materialien und Energie, Elektronenspeicherring BESSY II, Albert-Einstein-Straße 15, 12489 Berlin, Germany. [5]Synchrotron SOLEIL, L'Orme des Merisiers, Départementale 128, 91190 Saint-Aubin, France. [6]Department of Physics and Beijing Key Laboratory of Opto-electronic Functional Materials & Micro-nano Devices, Renmin University of China, Beijing 100872, China. [7]MAX IV Laboratory, Lund University, 22100 Lund, Sweden. [8]These authors contributed equally: Cong Li, Jianfeng Zhang, Yang Wang, Hongxiong Liu. ✉e-mail: conli@kth.se; oscar@kth.se

the magnetic transition temperature, the magnetic structure will affect the existing Weyl fermions as well as generate additional Weyl fermions. Therefore, the RAlX family provides an appropriate platform to study the interaction between magnetism and Weyl fermions. The interaction between magnetism and Weyl fermions includes two parts: the first part is the mediating effect of Weyl fermions on magnetism[57], and the second part is magnetism regulation of the Weyl fermions and further effects on the topological ordering. To date, the mediating effect of Weyl fermions on magnetism has been confirmed by neutron diffraction measurements[57]. However, unambiguous and direct experimental confirmation of the regulation of Weyl fermions by magnetism remains unobserved.

Here, we present angle-resolved photoemission spectroscopy (ARPES) measurements and band structure calculations to systematically investigate the electronic structure and topological properties of NdAlSi and how they are regulated by ferrimagnetism. We observe Fermi arcs and bulk Weyl fermion dispersion in the paramagnetic state, after determining a surface state associated with a Nd terminated surface cleaved at the Nd-Al plane. In addition, we observe a new Weyl fermion dispersion corresponding to the Weyl fermion generated in the ferrimagnetic state. These observations are in good agreement with the prediction of theoretical calculations and confirm the existence of Weyl fermions regulated by ferrimagnetism in NdAlSi. Our results provide key insights into the interplay between magnetism and topological orders.

NdAlSi is predicted to be a magnetic Weyl semimetal that crystallizes in the tetragonal structure with the space group $I4_1md$ (no. 109)[57], as shown in Fig. 1a. The crystal structure of NdAlSi has two vertical mirror planes, $m_x$ and $m_y$, as well as two vertical glide mirror planes, $m_{xy}$ and $m_{x\bar{y}}$, but lacks the mirror symmetry of the horizontal plane which induces the inversion symmetry breaking. The corresponding three-dimensional (3D) Brillouin zone (BZ) of NdAlSi is shown in Fig. 1b. In order to determine the magnetic structure of NdAlSi at low temperature, we performed magnetic susceptibility measurements (for details see Fig. S1 in the Supplemental Material). Based on the magnetic susceptibility (Fig. S1) and previous neutron scattering measurements[57], it can be inferred that the spontaneous magnetization of NdAlSi appears as a up-up-down ($\uparrow\uparrow\downarrow$, UUD) [or down-down-up ($\downarrow\downarrow\uparrow$, DDU)] type ferrimagnetism at low temperature ($T < T_{com} \sim 3.4$ K), as shown in Fig. 1c. The corresponding 3D BZ of NdAlSi in the UUD (or DDU) ferrimagnetic state is shown in Fig. 1d. In order to study the electronic structure and topological properties of NdAlSi and its regulation by ferrimagnetic order, we performed density functional theory (DFT) calculations in the paramagnetic and the UUD ferrimagnetic states, respectively. In the paramagnetic state, we found a total of 40 Weyl nodes whose locations within the first 3D BZ are shown in Fig. 1e, f. These Weyl points are mainly distributed in the $k_z \sim 0$ $\pi/c$ (Fig. 1g) and $k_z \sim \pm 0.67$ $\pi/c$ (Fig. 1h) planes. In the UUD ferrimagnetic state, the BZ is reconstructed relative to the paramagnetic state, resulting in the folding of the electronic structure and Weyl points. Figure 1i shows the distributions of all Weyl fermions in the

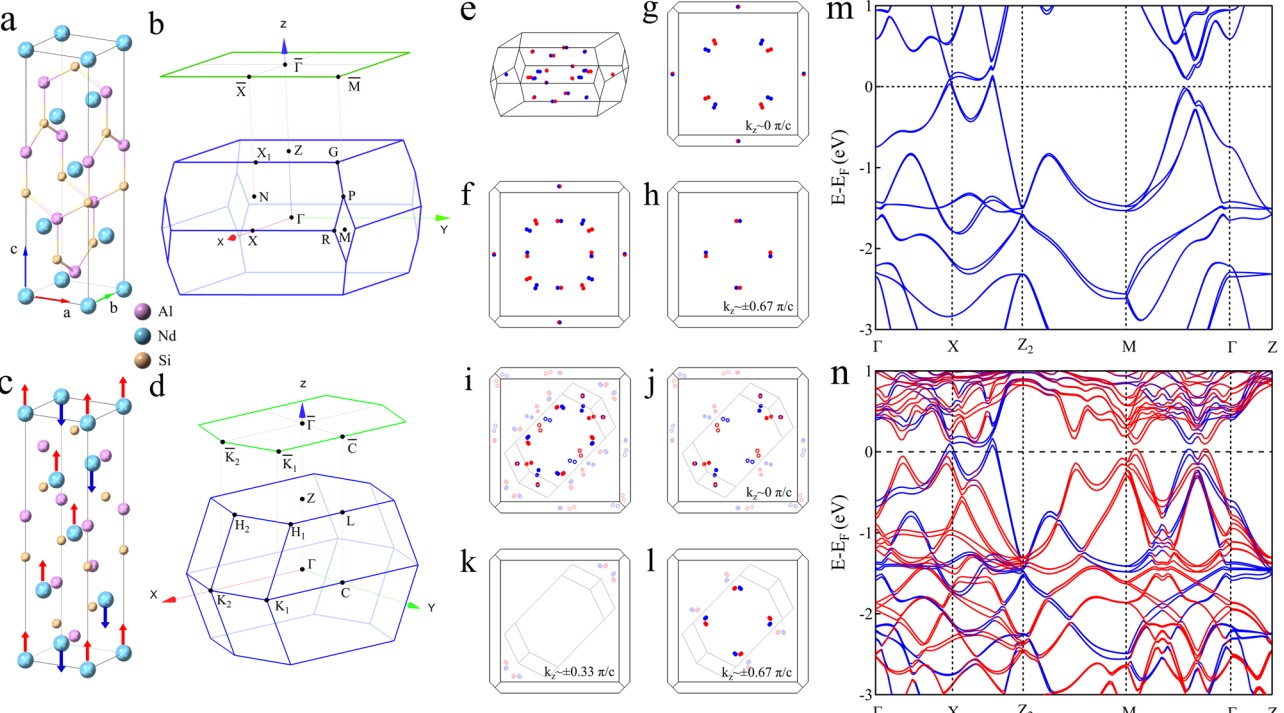

**Fig. 1 | Crystal structure and band structure calculations for NdAlSi. a** The crystal structure of NdAlSi. **b** The 3D BZ of the original unit cell of NdAlSi. **c** A three dimensional lattice view of the NdAlSi spin structure. **d** The corresponding 3D BZ of NdAlSi with the UUD ferrimagnetic order which exhibit the BZ folding relative to the paramagnetic state. **e** The distribution of Weyl fermions in the 3D BZ in the paramagnetic state, while the red dots representing nodes with chiralities +1 and blue dots representing -1. (**f**) The top view of (**e**). **g, h** The top view of (**e**) at the $k_z \sim 0$ $\pi/c$ plane (**g**) and - $\pm 0.67$ $\pi/c$ plane (**h**). **i** Distributions of all Weyl fermions from the top view of the BZ in ferrimagnetic state. The gray lines show the boundary of the BZ in the ferrimagnetic state. The bright blue and red dots represent the Weyl fermions in the folded BZ due to the ferrimagnetic order. The shaded blue and red dots are the Weyl fermions in the ferrimagnetic state after folding back into the first

BZ of the paramagnetic state (Weyl nodes folded back that are distributed in the - $\pm\pi/c$ plane are not shown here). **j–l** The top view of (**i**) at the $k_z \sim 0$ $\pi/c$ plane (**j**), - $\pm 0.33$ $\pi/c$ plane (**k**) and - $\pm 0.67$ $\pi/c$ plane (**l**). The new Weyl points resulting from BZ folding are represented by hollow dots. (**m, n**) Calculated band structures of NdAlSi along high-symmetry directions across the BZ in paramagnetic (**m**) and UUD type ferrimagnetic (**n**) state. The high symmetry points are defined in (**b**), while $Z_2$ is the Z point in the second BZ in the $k_z = 0$ $\pi/c$ plane. The blue lines are the bands corresponding to the paramagnetic state, and the red lines are the folded bands due to the UUD type ferrimagnetic order. Since the band calculation of DDU type ferrimagnetism is basically the same as that of UUD type ferrimagnetism (see Fig. S2 in the Supplemental Material), the UUD type ferrimagnetism is taken as an example in this paper.

ferrimagnetic state, to facilitate direct comparison with Weyl fermions in the paramagnetic state, we fold the Weyl fermions back into the first BZ in the paramagnetic state (gray red and blue dots in Fig. 1i). It is found that the folded back Weyl nodes are mainly distributed in the $k_z \sim 0$ $\pi/c$ (Fig. 1j), $k_z \sim \pm 0.33$ $\pi/c$ (Fig. 1k) and $k_z \sim \pm 0.67$ $\pi/c$ (Fig. 1l) planes of the first BZ of the paramagnetic state. Figure 1m,n show the DFT band structure calculations along $\Gamma$-X-$Z_2$-M-$\Gamma$-Z directions in the paramagnetic (Fig. 1m) and ferrimagnetic (Fig. 1n) states. This elucidates the influence of the UUD ferrimagnetic order on the electronic structure and topological properties of NdAlSi.

According to the above DFT calculations, the UUD ferrimagnetic order not only regulates the position of the existing Weyl nodes, but also generates new Weyl nodes in the 3D BZ. To confirm this prediction, we preformed ARPES measurements. We first study the electronic structure and topological properties of the paramagnetic state in detail, and then observe the influence of the UUD ferrimagnetic order induced by lowering the temperature.

Figure 2 displays the overall constant energy contours and band structures of NdAlSi measured by ARPES in the paramagnetic state. The evolution of constant energy contours of the electronic bands at different binding energies (Fig. 2a) exhibit sophisticated structures. A simple comparison with the DFT calculations of the bulk Fermi surface (Fig. S3 in the Supplemental Material) shows that the ARPES measurements (Fig. 2b) display a more complex structure, which may be related to the presence of surface states. In order to determine the contribution from surface states, we performed surface projected DFT

band calculations on six different possible (001) termination surfaces of NdAlSi (for the detailed band structure calculations see Fig. S4 in the Supplemental Material). The results show that the band structure calculations along the $\overline{Y}$-$\overline{\Gamma}$-$\overline{Y}$ (Fig. 2g) and $\overline{M}$-$\overline{\Gamma}$-$\overline{M}$ (Fig. 2h) directions on the Nd atom terminated surface of the Nd-Al cleavage plane can capture most of the measured band features in the corresponding direction (Fig. 2e, f) except the M shaped band at 0.8 eV binding energy and some very small details in Fig. 2e. When the two orthogonal domain structures leading to superposition of bands along the $\overline{X}$-$\overline{\Gamma}$-$\overline{X}$ and $\overline{Y}$-$\overline{\Gamma}$-$\overline{Y}$ directions are taken into account (Fig. 2i, j), all the band features can be understood. The surface projected DFT Fermi surface calculations on the Nd terminated surface of the Nd-Al cleavage plane is shown in Fig. 2c and agrees well with the measured constant energy contours (Fig. 2b) and when the two orthogonal domain structures are taken into account (Fig. 2d) it comfirms that the measured Fermi surface (Fig. 2b) is mainly from the surface states (for detailed analysis see the Supplemental Material).

We now proceed to explore Fermi arc candidates. By surface Green function calculations, we find two Fermi arc surface states connecting two pairs of Weyl nodes at the energy of 38 meV (Fig. 2k) and 56 meV (Fig. 2l) above the Fermi level (for details see the Supplemental Material). Similar arc features can also be found in the Fermi surface mapping in the same position, as shown in Fig. 2b. To confirm that the observed arc features in Fig. 2b are the Fermi arc surface states. We carried out photon energy dependent band structure measurements (Fig. 2m-p) at 15 K along the direction of

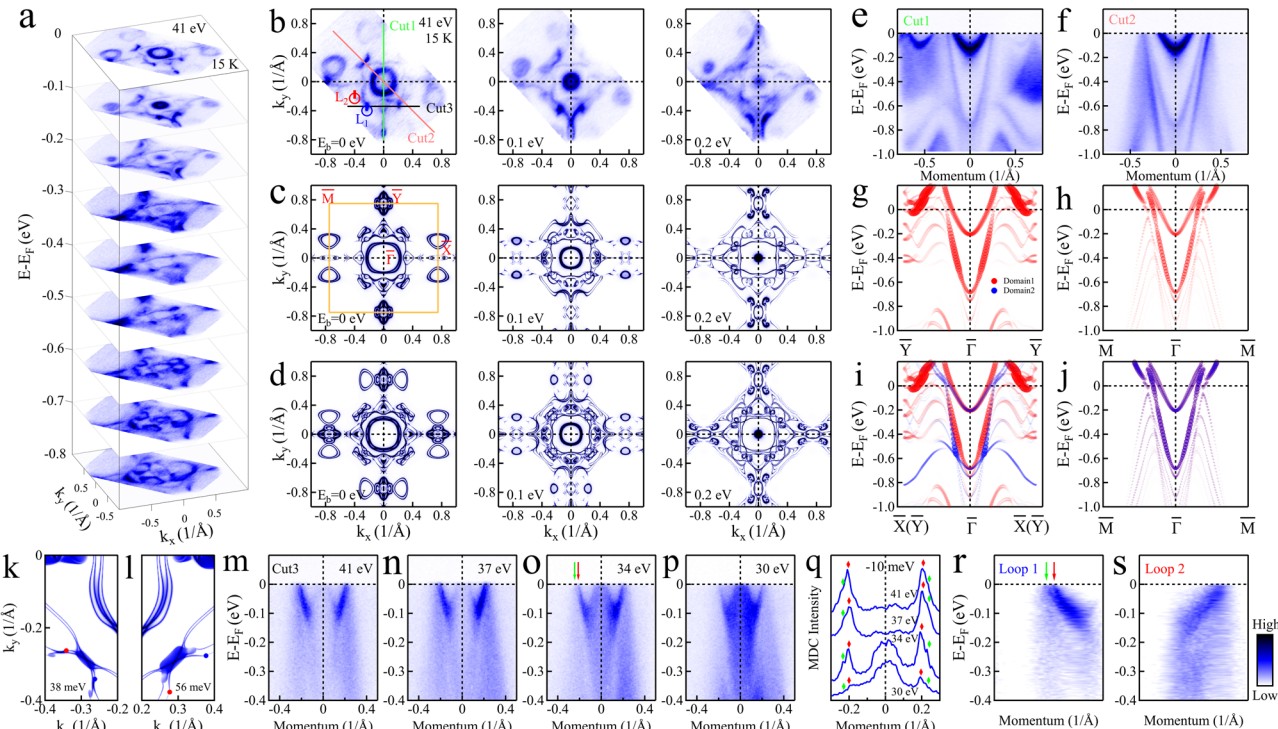

**Fig. 2 | Determination of the surface termination and observation of topological Fermi arcs and chiral charges in NdAlSi. a** Stacking plots of constant energy contours at different binding energies ($E_b$), obtained from ARPES, display complex band structure evolution as a function of energy. **b** Fermi surface and constant energy contours at different binding energies measured with a photon energy of 41 eV at 15 K under positive circular (PC) polarization. **c** Surface projected DFT calculations of the (001) Fermi surface and constant energy contours on the Nd atom terminated surface. **d** Surface projected DFT calculations for two domain structures derived from (**c**) and their superposition after rotation of 90 degrees. **e, f** Band dispersions along $\overline{Y}$-$\overline{\Gamma}$-$\overline{Y}$ [Cut1, (**e**)] and $\overline{M}$-$\overline{\Gamma}$-$\overline{M}$ [Cut2, (**f**)] directions. The momentum directions of the cuts are show in the leftmost panel of (**b**). **g, h** The

corresponding surface projected DFT calculations of the bands along $\overline{Y}$-$\overline{\Gamma}$-$\overline{Y}$ (**g**) and $\overline{M}$-$\overline{\Gamma}$-$\overline{M}$ (**h**) directions. (**i, j**) The surface projected DFT calculations of the bands along $\overline{Y}$-$\overline{\Gamma}$-$\overline{Y}$ (**i**) and $\overline{M}$-$\overline{\Gamma}$-$\overline{M}$ (**j**) directions, considering the two domain structures. The spectral intensity is illustrated by the size and transparency of the markers. **k, l** Surface Green function calculation of constant energy contours, taking into account the double domain structures, at an energy of 38 meV (**k**) and 56 meV (**l**) above the Fermi level. **m–p** Photon energy dependent band dispersions along Cut3 in (**b**). **q** The momentum distribution curves (MDCs) extracted from (**m–p**) at a binding energy of 10 meV. The red and green diamond shaped markers indicate the peaks positions corresponding to two different Fermi arcs. **r, s** Band dispersions along Loop 1 [L1, (**r**)] and Loop 2 [L2, (**s**)].

Cut3 ($k_y = 0.34$ Å$^{-1}$) in Fig. 2b. For further quantitative analysis, the extracted photon energy dependent momentum distribution curves (MDCs) at a binding energy of 10 meV from Fig. 2m-p are plotted in Fig. 2q. The MDCs exhibit negligible photon energy dependence, suggesting that it is a surface state. To further confirm that the arc features observed in Fig. 2b are indeed the Weyl Fermi arcs, we examine the signatures of chiral charge in NdAlSi based on the bulk-boundary correspondence between the bulk Weyl fermions and surface Fermi arcs[1,67]. In order to do so, band dispersion was measured along the straight Cut3 in Fig. 2b. Along this cut (Fig. 2m–p), a left-moving (Chern number $n_l = -1$) and right-moving (Chern numbe $n_r = +1$) edge mode related by a mirror plane $k_x = 0$ is clearly observed and associated with a 2D momentum-space slice carrying a total Chern number $n_{tot} = 0$. In addition, we also examine the signatures of chiral charge in the close loop cuts (Fig. 2b). For a closed loop in the surface BZ where the bulk band structure is everywhere gaped we add up the signs of the Fermi velocities of all surface states along this loop, with Chern number $n = +1$ for right movers and $n = -1$ for left movers[67]. Figure 2r, s show the unrolling of the closed loop cuts along Loop1 (Fig. 2r, L1 in Fig. 2b) and Loop2 (Fig. 2s, L2 in Fig. 2b). Loop L1 shows a left-moving chiral mode while a right-moving chiral mode is observed for loop L2. These results

unambiguously show that the arc features in Fig. 2b are the topological Fermi arcs.

After having identified the topological Fermi arc, we next search for the characteristic bulk Weyl fermion dispersion. Looking at the results from DFT calculations, we find a total of 40 Weyl nodes in the paramagnetic state, with 24 of them (W2, W3 and W4) distributed in the $k_z$ ~ 0 $\pi/c$ plane (Fig. 3c) and the others (W1) are situated in the $k_z$ ~ ± 0.67 $\pi/c$ plane (Fig. 3b). In order to precisely identify the $k_z$ momentum locations of the Weyl nodes, we performed broad-range (30 to 90 eV) photon energy dependent ARPES measurements on a cleaved sample with a poor surface so that the surface states are completely suppressed (for details see Supplemental Material). Due to the absence of interference from surface states, bulk bands can be clearly distinguished and characterized. These results were further confirmed using bulk sensitive soft X-ray ARPES (SX-ARPES) measurements (for details see Supplemental Material). Figure 3a shows the photon energy dependent ARPES spectral intensity map ($k_x$-$k_z$ Fermi surface) at a binding energy of 0.2 eV along the $\overline{X}$-$\overline{\Gamma}$-$\overline{X}$ direction, measured at 30 K. By analyzing the periodic structure along the $k_z$ direction, the correspondence between the high symmetry points of the BZ along the $k_z$ direction and the photon energy is determined as shown in Fig. 3a. Based on this, the $k_z$ ~ ± 0.67 $\pi/c$ plane hosting W1 type

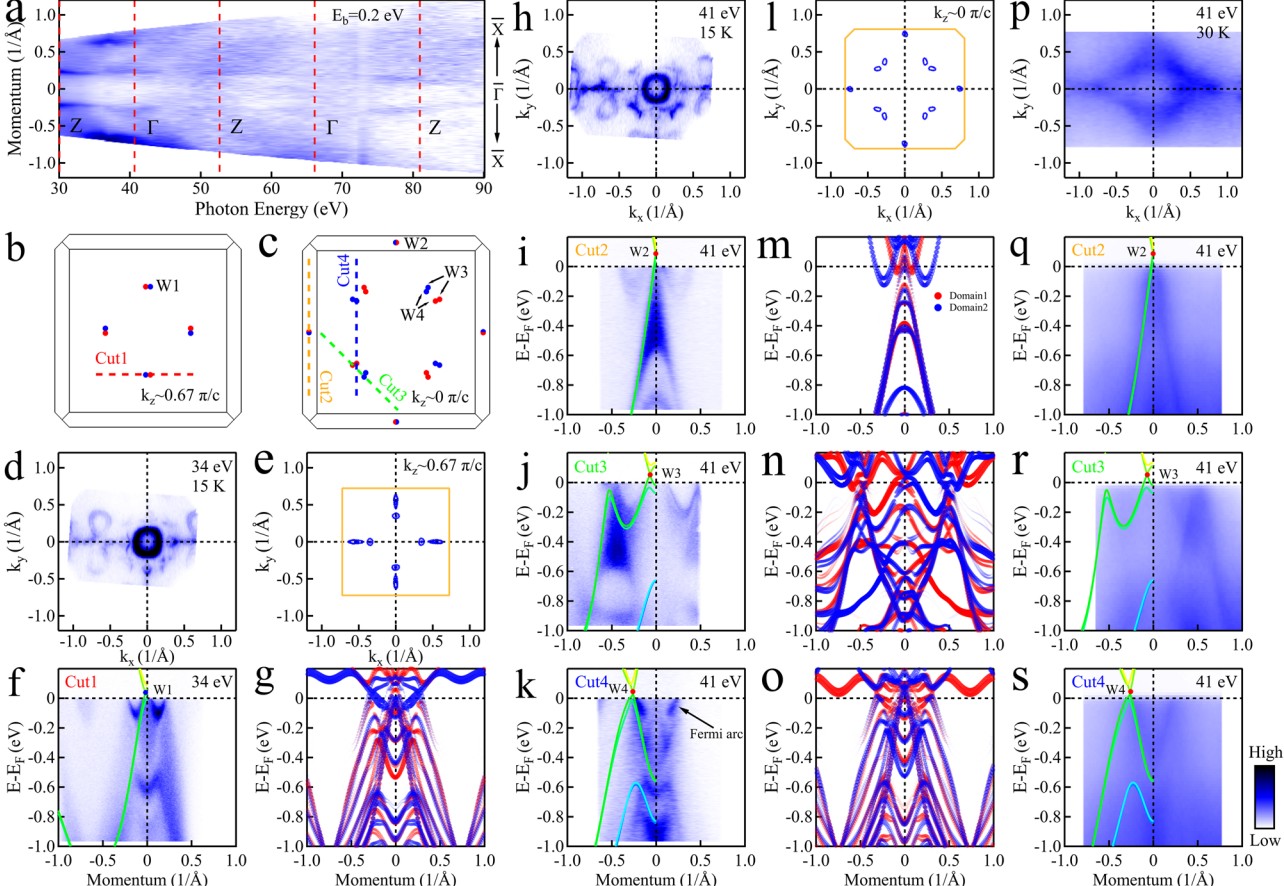

**Fig. 3 | Weyl fermions in NdAlSi. a** Photon energy dependent ARPES spectral intensity map at the binding energy of 0.2 eV along the $\overline{X}$-$\overline{\Gamma}$-$\overline{X}$ direction measured at 30 K. For the sake of comparison, the spectral intensities are normalized. **b, c** The distribution of Weyl fermions in the BZ at the $k_z$ - 0.67 $\pi/c$ plane (**b**) and -0 $\pi/c$ plane (**c**) in which the paths of Cut1 to Cut4 and the types of Weyl fermion W1, W2, W3 and W4 are defined. **d** The Fermi surface map measured at 15 K with photon energy of 34 eV under PC polarization. **e** Calculated bulk Fermi surface at the $k_z$ - 0.67 $\pi/c$ plane. The BZ is defined by orange solid lines. **f** Band dispersion along Cut1 measured with a photon energy of 34 eV at 15 K related to Weyl fermions W1, with DFT bulk calculations overlaid. **g** Corresponding surface projected DFT calculations for

(**f**). The spectral intensity is denoted by the size and transparency of the markers. (**h**) The Fermi surface map measured at 15 K with a photon energy of 41 eV under PC polarization. **i–k** Band dispersion measured along Cut2 to Cut4 with a photon energy of 41 eV under PC polarization at 15 K related to Weyl fermions W2 (**i**), W3 (**j**) and W4 (**k**), with DFT bulk calculations overlaid. **l** Calculated bulk Fermi surface at the $k_z$ - 0 $\pi/c$ plane. The BZ is defined by orange solid lines. **m–o** Corresponding surface projected DFT calculations for (**i–k**). **p** The Fermi surface map measured on a low quality surface at 30 K with a photon energy of 41 eV under LH polarization. **q–s** Similar measurements as (**i–k**) but measured on the low quality surface sample at 30 K with photon energy of 41 eV under LH polarization.

of Weyl fermions corresponds to a photon energy of ~34 eV and the $k_z \sim 0$ $\pi/c$ plane hosting W2, W3 and W4 types of Weyl fermion corresponds to a photon energy of ~41 eV.

Figure 3 d, h show the Fermi surface mapping of NdAlSi measured at 15 K with the photon energy of 34 eV (Fig. 3d) and 41 eV (Fig. 3h) under Positive Circular (PC) polarization on a high quality cleaved surface. The experimental data and the calculated bulk Fermi surface (Fig. 3e, l) are distinctively different due to the admixture of the surface states. The Fermi surface mapping of the low quality surface (Fig. 3p), on the other hand, shows good overall agreement with the bulk calculations (Fig. 3l) due to the suppression of the surface states. In order to expose the Weyl fermion dispersion, we conducted band dispersion measurements along 4 cuts (Fig. 3b, c). The corresponding dispersion data is displayed for cuts 1-4 in Fig. 3f, i–k for the high quality surface and for cuts 2-4 in Fig. 3q–s for the low quality surface. The corresponding surface projected DFT calculations are plotted in Fig. 3g,m-o for comparison with the corresponding DFT bulk calculations overlayed. It is observed that the measured dispersions on the low quality surface (Fig. 3q–s) show good overall agreement with DFT bulk calculations and the measured dispersions on the high quality surface (Fig. 3f, i–k) are also consistent with the surface projected DFT calculations (Fig. 3g, m–o). The observation of the topological Fermi arc and bulk Weyl fermions with linear dispersions, as well as the consistency between theoretical calculations and experimental observations, establishes NdAlSi as a Weyl semimetal.

With the topological properties of the paramagnetic state well established, we turn to the question of how these topological properties are regulated by the ferrimagnetism. The ferrimagnetism in NdAlSi comes from the f orbital of Nd atoms, and the interaction between Weyl fermions and magnetism is mainly of Ruderman-Kittel-Kasuya-Yosida (RKKY) type, the magnitude of which oscillates with the distance between the Nd atoms[68]. The results from the DFT bulk calculations show that the effect of ferrimagnetism on Weyl fermion is mainly reflected in two parts. The first part is that the net magnetic moment causes the TRS breaking and drives a shift of the Weyl points in the 3D BZ, which is also expected to be observed in CeAlSi[43,46]. However, the actual shift of the Weyl point in 3D BZ is not easy to be measured according to the current accuracy. The second part is that the ferrimagnetic order leads to the folding of the BZ and generates new Weyl points in the BZ relative to the paramagnetic state. Based on the DFT calculations, in the ferrimagnetic state, if the Weyl fermions are folded back into the first BZ of the paramagnetic state, the back folded Weyl nodes are mainly distributed in the $k_z \sim 0$ $\pi/c$ (Fig. 4b), $k_z \sim \pm 0.33$ $\pi/c$ (Fig. 4c) and $k_z \sim \pm 0.67$ $\pi/c$ (Fig. 4d) planes of the first BZ of the paramagnetic state. By comparing the distribution of Weyl fermions to the paramagnetic state, it can be found that the folding of the BZ caused by ferrimagnetism generates 4 pairs of W2, 2 pairs of W3 and 2 pairs of W4 type Weyl fermions in the $k_z \sim 0$ $\pi/c$ plane (Fig. 4b) as well as 4 pairs of W1 type of Weyl fermions in the $k_z \sim \pm 0.67$ $\pi/c$ plane (Fig. 4d) of the first BZ of the UUD ferrimagnetic state. Furthermore, 4 pairs of W1 type of Weyl fermions are generated in the $k_z \sim \pm 0.33$ $\pi/c$ plane (Fig. 4c) of the first BZ of the paramagnetic state. Therefore, we can use the generation of the Weyl fermions mentioned above as a criterion to judge the presence of Weyl fermions regulated by the UUD

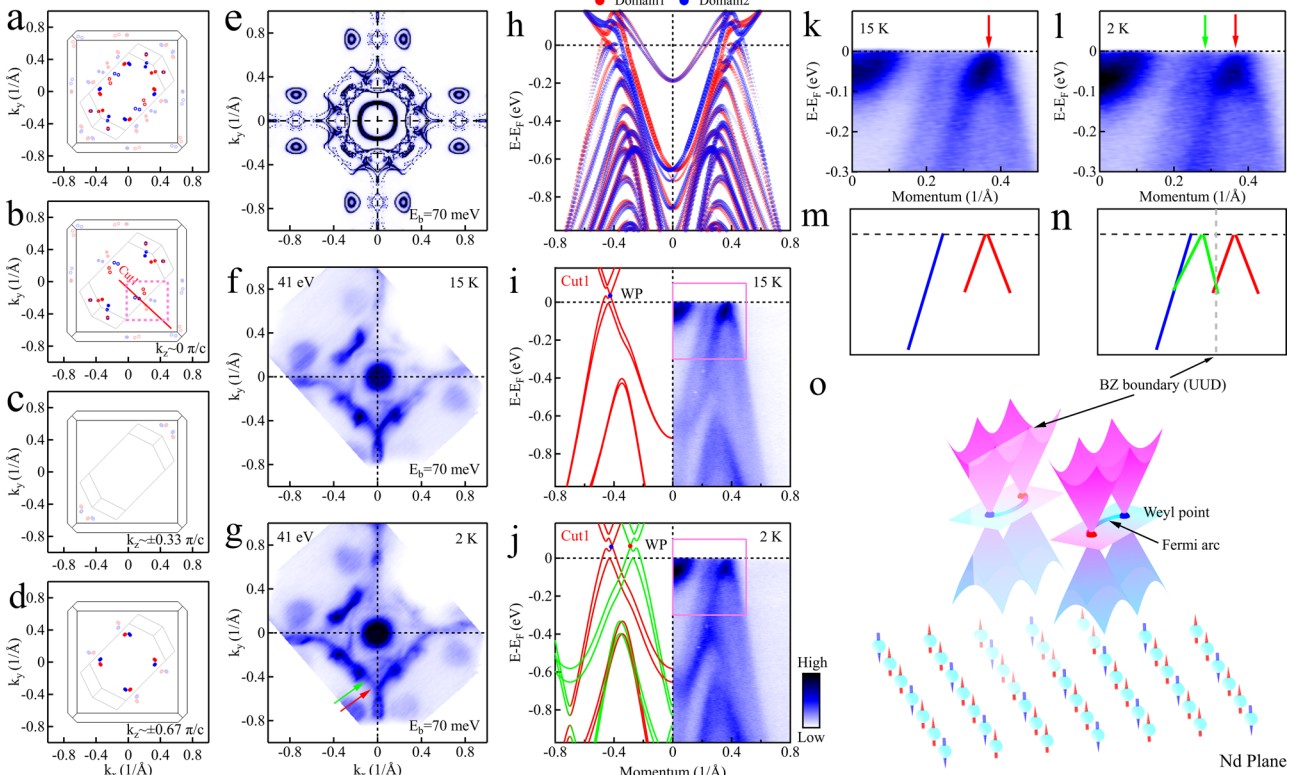

**Fig. 4 | Ferrimagnetic regulation of Weyl fermions in NdAlSi.** For the convenience of description, the distributions of Weyl fermions from the top view of the BZ in ferrimagnetic state are replotted in (**a–d**). **e** Surface projected DFT calculations of constant energy contours at the binding energy of 70 meV in paramagnetic state. **f, g** The constant energy contours at the binding energy of 70 meV measured with a photon energy of 41 eV at 15 K (**f**) and 2 K (**g**). **h** The surface projected DFT calculations of the bands in the paramagnetic state along the path of Cut1 in (**b**). The spectral intensity is denoted by the size and transparency of the markers. **i, j** The band dispersions along the path of Cut1 in (**b**) measured with photon energy of 41 eV under PC polarization at 15 K (**i**) and 2 K (**j**), with DFT bulk calculations overlaid on the left side. **k, l** Zoom in on the areas in (**i, j**) enclosed by the pink lines. **m, n** The band structure diagrams extracted from (**k**) and (**l**) in paramagnetic (**m**) and ferrimagnetic (**n**) states, respectively. **o** Cartoon picture of the regulation of Weyl fermions by ferrimagnetism in NdAlSi in the $k_z \sim 0$ $\pi/c$ plane, corresponding to the area enclosed by dashed pink lines in (**b**).

ferrimagnetic order. Figure 4f, g show the constant energy contours at the binding energy of 70 meV measured with a photon energy of 41 eV at 15 K (Fig. 4f) and 2K ($T < T_{com}$ ~ 3.4 K) (Fig. 4g). As the temperature is lowered below $T_{com}$, a new feature (marked by a green arrow in Fig. 4g) can be observed. This feature is not present at 15 K (Fig. 4f) as well as the surface projected DFT calculations in paramagnetic state (Fig. 4e). First, we conclude that the new feature is unlikely to be caused by surface reconstruction, although surface reconstruction has previously been reported to produce or suppress some Fermi surface features[69]. Since no structural phase transitions have been reported except for magnetic phase transitions in the temperature range from 2 K to 15 K in NdAlSi. The probability of spontaneous surface reconstruction occurring in the temperature range from 2 K to 15 K and forming a triple charge density wave (CDW) exactly along the in-plane diagonal direction should be very small. In terms of the feature shape, it is mirror-symmetric to the features marked by the red arrow and can be related to the folding of the electronic structure caused by the UUD ferrimagnetic order (green lines in Fig. 4j).To further confirm this conjecture, we conducted temperature dependent band dispersion measurements (Fig. 4i–l) along Cut1 in Fig. 4b, and compared them to the corresponding surface projected DFT band calculations in the paramagnetic state (Fig. 4h). All the band features measured at 15 K (Fig. 4i) are captured by the band calculations in the paramagnetic state (Fig. 4h). However, the features marked by a green arrow in Fig. 4l can not be found in Fig. 4k. It appears to be a duplicate of the band marked by the red arrow in Fig. 4l (for detailed MDCs analysis see Fig. S11 in the Supplemental Material). The noteworthy feature is that they are mirror-symmetric exactly around the boundary of the BZ of the ferrimagnetic state and break at 15 K (for more comparison of band dispersion cuts see Fig. S12 in the Supplemental Material). Therefore, we infer that the new features generated at 2 K are the folded electronic structures caused by ferrimagnetism (green lines in Fig. 4n). It also implies that new Weyl fermions are generated in conjunction with the appearance of the UUD ferrimagnetism. A cartoon picture of this process is shown in Fig. 4o.

The present observations establish NdAlSi as a ferrimagnetic Weyl semimetal and provide strong evidence for the interaction between ferrimagnetism and Weyl fermions. This helps us understand why there is an incommensurate ferrimagnetic order mediated by Weyl fermions between 3.4 K and 7.3 K[57]. In the ferrimagnetic state, ferrimagnetism not only drives Weyl fermion shifting in the 3D BZ due to the breaking of TRS, but also generate new Weyl fermions. This, in turn, results in an increase of the density of states near the Fermi level, leading to higher conductivity. The expected increase in conductivity is consistent with transport measurements[57,58]. In addition, new Weyl fermions generated in the ferrimagnetic state can also impact the thermal conductivity of ferrimagnetic Weyl semimetals. The increasing number of the Weyl nodes and Fermi arcs in the 3D BZ maybe lead to a reduction in the thermal conductivity due to the increased scattering of heat-carrying phonons. This reduction in thermal conductivity can result in improved thermoelectric performance, making ferrimagnetic Weyl semimetals potentially useful for energy conversion applications. Heat transport measurements are needed to confirm this.

In summary, ARPES measurements combined with band structure calculations paint a clear physical picture of the ferrimagnetic regulation of Weyl fermions. This interplay is key to understanding the electrical and thermal transport properties of ferrimagnetic Weyl semimetals and could facilitate the development of new technologies based on these materials. Our study provides spectral evidence for the interplay between magnetism and Weyl fermions in condensed matter physics and opens up a new avenue to study the magnetic regulation of topological ordering, a topic of great interest in topological physics.

## Methods

### Sample
Single crystals of NdAlSi were grown from Al as flux. Nd, Al, Si elements were sealed in an alumina crucible with the molar ratio of 1: 10: 1. The crucible was finally sealed in a highly evacuated quartz tube. The tube was heated up to 1273 K, maintained for 12 h and then cooled down to 973 K at a rate of 3 K per hour. Single crystals were separated from the flux by centrifuging. The Al flux attached to the single crystals were removed by dilute NaOH solution.

### ARPES measurements
High-resolution ARPES measurements were performed at the beamline UE112 PGM-2b-1[3] at BESSY II (Berlin Electron Storage Ring Society for Synchrotron Radiation) synchrotron, equipped with a SCIENTA R8000+DA30L electron analyzer and at the CASSIOPEE beamline of the SOLEIL synchrotron, equipped with a SCIENTA R4000 electron analyzer. The total energy resolution (analyzer and beamline) was set to 5‐20 meV for the measurements. The angular resolution of the analyser was ~ 0.1 degree. The samples were cleaved in situ and measured at different temperatures in ultrahigh vacuum with a base pressure better than $1.0 \times 10^{-10}$ mbar. Soft X-ray (SX) ARPES measurements were performed at the VERITAS beamline of the MAX IV synchrotron equipped with a R4000 electron analyzer.

### DFT calculations
The electronic structure calculations for NdAlSi were performed based on the density functional theory (DFT)[70,71] as implemented in the VASP package[72,73]. The generalized gradient approximation (GGA) of Perdew-Burke-Ernzerhof (PBE) type[74] was chosen for the exchange-correlation functional. The projector augmented wave (PAW) method[75,76] was adopted to describe the interactions between valence electrons and nuclei. In the calculation of the high-temperature paramagnetic phase of NdAlSi, the Nd pseudopotential was chosen without the 4$f$ electrons. The kinetic energy cutoff of the plane-wave basis was set to be 350 eV. A $16 \times 16 \times 16$ Monkhorst-Pack grids[77] was used for the BZ sampling. For describing the Fermi-Dirac distribution function, a Gaussian smearing of 0.05 eV was used. For structure optimization, both lattice parameters and internal atomic positions were fully relaxed until the forces on all atoms were <0.01 eV/Å. The relaxed lattice constants are: $a_0$ = 4.22 Å and $c_0$ = 14.59 Å, which agree well with the experimental measurements[58]. The spin-orbit coupling effect was also included for studying the non-trivial band topological. The positions and chirality of the Weyl points were studied by using Wannier-Tools package[78], which interfaces the Wannier90 package[79]. For further studying the surface states of NdAlSi with different terminal surfaces, we employed a 36 atomic layers slab system with 20 Å vacuum. The orbital weights of the top three atomic layers were calculated as the surface states with respect to the corresponding terminal surfaces.

As for the calculation of the low-temperature ferrimagnetic phase, we adopted the pseudopotential of Nd including the 4$f$ electrons. The kinetic energy cutoff of the plane-wave basis was set to be 450 eV. A ferrimagnetic supercell was constructed the same with ref. 57, and a $6 \times 6 \times 6$ Monkhorst-Pack grids was used for its folded BZ sampling. The electronic correlation effect among Nd 4$f$ electrons was incorporated by using the GGA+U framework of Dudarev formalism[80]. The effective U value on Nd 4$f$ electrons was set to be 6 eV, which is the same with the previous study[57]. For checking the influence of the FIM state on the Weyl point distribution around the Fermi level, we performed band unfolding calculations by using the VASPKIT package[81].

## Data availability
The authors declare that all data supporting the findings of this study are available within the paper and its Supplementary Information files.

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

## Acknowledgements

The work presented here was financially supported by the Swedish Research council (2019-00701) and the Knut and Alice Wallenberg foundation (2018.0104). M.D. acknowledges financial support from the Göran Gustafsson Foundation and Swedish Research Council (2022-03813). D.P. acknowledges Swedish Research Council (2020-00681). Y.G.S. acknowledges the National Natural Science Foundation of China (Grants No. U2032204), and the Informatization Plan of Chinese Academy of Sciences (CAS-WX2021SF-0102). Research conducted at MAX IV, a Swedish national user facility, is supported by the Swedish Research council under contract 2018-07152, the Swedish Governmental Agency for Innovation Systems under contract 2018-04969, and Formas under contract 2019-02496.

## Author contributions

C.L. proposed and conceived the project. C.L. carried out the ARPES experiments with the assistance from Y.W., Q.D.G., W.Y.C. and D.P.. J.F.Z., H.C.Y. and T.X. contributed to the band structure calculations and theoretical discussion. H.X.L. and Y.G.S. contributed to NdAlSi crystal growth. C.L. characterized the single crystals. C.L. contributed to software development for data analysis and analyzed the data with the assistance from Y.W.. C.L. wrote the paper. E.R., F.B., H.F. and B.T. provided the beamline support. M.D., M.H.B. and O.T. contributed to the scientific discussions. All authors participated in and commented on the paper.

## Funding

## Competing interests

The authors declare no competing interests.
