## [Peer Review File · Nature Communications]

Emergence of Weyl Fermions by Ferrimagnetism in a
Noncentrosymmetric Magnetic Weyl SemimetalReviewers' Comments:

Reviewer #1:

Remarks to the Author:

The authors systematically analyzed the topological orders of the paramagnetic state and the ferrimagnetic state in NdAlSi, respectively. With both experimental observation and theoretical calculations, NdAlSi has been established as a magnetic Weyl semimetal. And the ferrimagnetic regulation of Weyl fermions is studied in this work. But I don't think it is so important to find a new magnetic topological material again since there are already some reports about topological electronic band structures in recent years. So, I don't think this work is novel enough to meet the level of nature communications.

In addition, I have some questions that are not very clear.

1. In Fig. 1(k), it seems no Weyl fermions exist in the folded BZ of the ferrimagnetic state, why are there Weyl fermions after folding back into the first BZ of the paramagnetic state?
2. It is claimed in the manuscript that ferrimagnetism regulates the Weyl fermions in NdAlSi, but I don't see much analysis about this regulation, what is the mechanism of this regulation from the theoretical perspective?
3. The surface projected dispersions of the paramagnetic state along cut1 are shown in Fig. 4(d), which has been compared with experimental results in Fig. 4(e-f), but I am confused with the magnetic state of the measured constant energy surfaces and dispersions, and the authors should give the results of the surface projected dispersions of ferrimagnetic state.

Reviewer #2:

Remarks to the Author:

The manuscript by Li et.al. presents a comprehensive study of the electronic bandstructure of NdAlSi, a compound which has been recently discussed as a magnetic Weyl semimetal candidate. The study aims to unveil a possible regulation of Weyl fermions by the emergence of a magnetic ground state below a critical temperature. While the authors do an excellent job in mapping the electronic bandstructure and surface states of their in-situ cleaved NdAlSi crystals -strongly suggesting the existence of a Weyl Semimetal phase by carrying out well-known ARPES protocols-, the evidence for a "regulation" of Weyl fermions through a ferrimagnetic ground state is not clear from their experiments. From the electronic band calculations, when the crystal is in the ferrimagnetic state, there is an effect of band folding and consequently the natural creation of additional Weyl points visible in the folded BZ. In addition, the authors observe an increase of the spectral weight in one of the energy cuts while going through the magnetic transition temperature, which they attribute to the magnetic interactions in the crystal. However, these findings are too incipient to argue for a "regulation of Weyl Fermions" by magnetic ground states. The connection between magnetism and Weyl physics has to be further elaborated to support the claims. Therefore, the authors should either look for more convincing evidence of an active interplay between magnetism and Weyl fermions, or reformulate/revisit their main claims putting more emphasis on the bandstructure study, where they did a good job. I cannot recommend publication of the article in the present form, but I think there is room for improvement. In particular, through revision of the following major issues:

- 1) The first evidence of bandstructure changes by magnetism is provided by calculations of the UUD ferrimagnetic state of NdAlSi. Due to the band-folding effect, it is not straightforward to judge whether there is an emergence of new Weyl points, or just a redistribution of the WPs at different k space locations caused by band folding. The authors should mark in the constant energy contours in Fig. 1 (g-l) which new WPs are expected to appear as a result of magnetic ordering, besides the ones projected on the folded BZ. In the manuscript there is always a notion of "generation of new Weyl points", which should not be confused by projecting the existing WPs on other k locations in the BZ.
- 2) The physics underlying the effect of magnetic ordering in the bandstructure of NdAlSi. In magnetic solids, the most notable effect from a magnetic ground state is the spin-splitting of electronic bands, and in some cases, a shift of the degeneracy points in momentum space can be inferred (e.g. a recent

paper by Piva et al. on another WSM candidate CeAlSi, 10.1103/PhysRevResearch.5.013068). The authors should describe in detail what are the crucial factors that are responsible for a change in the bandstructure (i.e. band folding) in the ferrimagnetic phase: i) the mere fact that you have exchange interactions between the Nd atoms; ii) a potential change in atomic bond lengths, iii) any type of surface reconstruction that may happen at low temperatures. A surface reconstruction has been shown (see. e.g. *Advanced Materials* 33 (21), 2008634, 2021) to induce band folding and create "new" Weyl features on the folded bandstructure). The authors should refer to this publication and discuss other origins of band folding other than magnetic interactions.

3) In conjunction with the first two points, a figure comparing the experimental k_x - k_y contour (e.g. Fig. 4b) at the same binding energy and photon energy of the paramagnetic and ferrimagnetic UUD phase should be constructed together with the calculated contours at the same energies, to see more clearly the fingerprints of band folding in the spectra (not easy to compare the symmetry of the BZ in the experimental contours).

4) In several passages of the manuscript it is mentioned that new Weyl points are created in the UUD phase, but in Fig. 4c the additional weak feature cannot be safely attributed to be a Weyl point. The authors should make an effort to corroborate this important information, either by refining the measurements (Fig. 4h is just a blurry contrast with no structure to match with the green line) or by looking at other relevant energies/momenta where this effect also takes place.

5) In the present form, the title is misleading and oversells the content of the manuscript. Even in the best case scenario, where band folding and new Weyl points can be unambiguously ascribed to the ferrimagnetic phase, the "regulation of Weyl Fermions" is still far fetched. "Emergence of Weyl points in the ferrimagnetic phase of NdAlSi" or something along those lines will be more appropriate.

Minor issues:

1) Introduction: For a more general reading audience, it should be briefly commented why breaking of both TRS and IS still holds a Weyl Semimetal phase. Usually it is one or the other, not both at the same time.

2) The authors present a "low quality surface" measurement to suppress the surface states and just be sensitive to the bulk electronic dispersion. It would be instructive for the surface science community to describe how they prepare this low quality surface, whether it is a cleavage effect or rather a post-cleavage surface treatment in vacuum. Alternatively one can distinguish bulk and surface states by performing photon energy dependent measurements. I would be curious if the bulk states (and WP dispersions) can be also corroborated that way.

3) Figure 1 (calculations) and Figure 4 (experiments) contain important information about the electronic structure of the ferrimagnetic state, which is the novelty of the manuscript. It is difficult for the reader to jump back and forth to follow the content, so the idea of first discussing the PM phase and then only the effect of the FM is more convenient for the manuscript outline (and one avoids repetitive content at the discussions of Fig. 1 and 4). Especially for major issue 3) putting the calculated k_x - k_y next to the spectra of Figs 4b and 4c will help a lot to identify the relevance of band folding and the new electronic features.

Dr. Amilcar Bedoya-Pinto

Reviewer #3:

Remarks to the Author:

Referee Report for NCOMMS-23-17139-T, "Ferrimagnetic Regulation of Weyl Fermions in a Noncentrosymmetric Magnetic Weyl Semimetal" by C. Li et al., is being considered for publication in *Nature Communication*.

This ARPES work is a tour de force with parts of the data taken at 2K. The manuscript aims to address the interplay between magnetism and topological orders in the rare earth RAlX (R: Rare earth; X: Si or Ge) family, with a specific focus on NdAlSi as a magnetic Weyl semimetal candidate, and the existence of Weyl fermions regulated by ferrimagnetism. The paper provides a systematic investigation into the electronic structure and topological properties of NdAlSi using angle-resolved photoemission spectroscopy (ARPES) measurements and density functional theory (DFT) calculations, with implications for the development of new materials for spintronics applications.

The paper was written in a very pedagogical and concise manner, making it an enjoyable read.

However, it was not clear from the text if ferrimagnetism plays any role different from ferromagnetism in Weyl semimetals. The authors pointed out in page 4 that the UUD ferrimagnetic phase would reconstruct the paramagnetic BZ, leading to folding of bands and creation of new Weyl nodes. I suggest the authors emphasize its contrast to ferromagnetic Weyl semimetals, such as PrAlGe and CeAlGe, where the onset of magnetism would only shift the Weyl nodes in k-space and split the bands due to Zeeman effect.

One of the biggest challenges for measuring magnetic materials with ARPES is finding a single domain. Typically, the magnetic domains are micron-size and therefore smaller than the beam spot size in most ARPES setups. Unless being field cooled, the sample is likely in a mixed state with UUD and DDU domains at 2K. However, Fig.4 seems to imply that the ARPES measurements at 2K were taken from a dominating magnetic domain. Could the authors please clarify this in the manuscript?

Figure 4 is quite confusing in comparing the dispersions measured at 15K vs 2K. The DFT overlay in Fig.4e is clearly not matching the data. The bands overlaid by the green lines in Fig.4f are also present in Fig.4e, but they were not captured by the DFT calculation. Comparing with surface projected DFT in Fig.4d, it seems to me that this "green-line band" is a surface state. However, looking at the raw data, I would argue that Fig.4e and Fig.4f are almost identical. It seems to me that the band overlaid by the green lines could also be equally well explained by a surface state. Even in the zoomed in figures, I find it hard to difficult to see the second set of bands under the green dashed lines. Could the authors please clarify? Perhaps second derivatives to the raw data, or a change of color scale would help?

The center electron pocket in Fig.4 has "filled up" intensity in the FS plot, instead of a closed loop. This is typically due to blurring from the k_z dispersion. However, it is clear from the surface projected DFT calculation that this electron pocket is a surface state. I also see that the other bands are reasonably sharp, and therefore the blurring cannot come from lack of surface quality. This is even clearer in Fig.S9, where the center electron pocket is much broader than the other bands. Do the authors understand why the blurring of intensity only happens to this electron pocket?

In summary, I find the current work highly significant and well written. However, I have my reservations about the ARPES features the authors ascribed to the ferrimagnetic onset. In particular, the generation of new Weyl points in the ferrimagnetic phase is not clear to me. I would recommend another round of discussion before making a decision on publication.

Response to Reviewer's Comments

Reviewer #1 Comments:

The authors systematically analyzed the topological orders of the paramagnetic state and the ferrimagnetic state in NdAlSi, respectively. With both experimental observation and theoretical calculations, NdAlSi has been established as a magnetic Weyl semimetal. And the ferrimagnetic regulation of Weyl fermions is studied in this work. But I don't think it is so important to find a new magnetic topological material again since there are already some reports about topological electronic band structures in recent years. So, I don't think this work is novel enough to meet the level of nature communications.

In addition, I have some questions that are not very clear.

1. In Fig. 1(k), it seems no Weyl fermions exist in the folded BZ of the ferrimagnetic state, why are there Weyl fermions after folding back into the first BZ of the paramagnetic state?

2. It is claimed in the manuscript that ferrimagnetism regulates the Weyl fermions in NdAlSi, but I don't see much analysis about this regulation, what is the mechanism of this regulation from the theoretical perspective?

3. The surface projected dispersions of the paramagnetic state along cut1 are shown in Fig. 4(d), which has been compared with experimental results in Fig. 4(e-f), but I am confused with the magnetic state of the measured constant energy surfaces and dispersions, and the authors should give the results of the surface projected dispersions of ferrimagnetic state.

Response to Reviewer #1

The authors systematically analyzed the topological orders of the paramagnetic state and the ferrimagnetic state in NdAlSi, respectively. With both experimental observation and theoretical calculations, NdAlSi has been established as a magnetic Weyl semimetal. And the ferrimagnetic regulation of Weyl fermions is studied in this work. But I don't think it is so important to find a new magnetic topological material again since there are already some reports about topological

electronic band structures in recent years. So, I don't think this work is novel enough to meet the level of nature communications.

We thank Reviewer #1 for the review and constructive comments. Unfortunately, we have the impression that Reviewer #1 may have misunderstood the main point of our work. The difference from the other magnetic topological materials discovered so far, breaking only time-reversal symmetry (TRS), is that NdAlSi is a rare case of a magnetic Weyl semimetal candidate that displays both inversion symmetry (IS) and TRS breaking with an incommensurate ferrimagnetic transition at 7.3 K and a commensurate ferrimagnetic transition at 3.4 K. Above the ferrimagnetic transition temperature, NdAlSi is already a Weyl semi-metal with 40 Weyl points in the first Brillouin zone (BZ) due to the IS breaking. Therefore, it provides a rare system appropriate for studying the interaction between magnetism and Weyl fermions. In this paper, we provide the first spectral evidence for the interplay between magnetism and Weyl fermions in condensed matter physics which opens up a new avenue to study the magnetic regulation of topological ordering, a topic of great interest in topological physics.

In the following, we give our response to the Reviewer's comments one by one.

- 1. In Fig. 1(k), it seems no Weyl fermions exist in the folded BZ of the ferrimagnetic state, why are there Weyl fermions after folding back into the first BZ of the paramagnetic state?*

The Weyl fermions after folding back into the first BZ of the paramagnetic state at $k_z \sim \pm 0.33 \pi/c$ plane are from the Weyl fermions of $k_z \sim \pm 0.67 \pi/c$ plane in the folded BZ of the ferrimagnetic state, as shown in Fig. R1_1. The bright blue and red dots represent the Weyl fermions in the folded BZ due to the ferrimagnetic order. The shaded blue and red dots are the Weyl fermions in the ferrimagnetic state after folding back into the first BZ of the paramagnetic state.

Fig. R1_1. The distribution of Weyl fermions in the folded BZ of the ferrimagnetic state

- 2. It is claimed in the manuscript that ferrimagnetism regulates the Weyl fermions in NdAlSi, but I don't see much analysis about this regulation, what is the mechanism of this regulation from the theoretical perspective?*

In NdAlSi, due to the presence of ferrimagnetism, a net magnetic moment is generated in the materials, which breaks the time reversal symmetry, and will drive the Weyl points shifting in the 3D BZ. In addition, the ferrimagnetic order can also cause the folding of the BZ and create new Weyl points in the BZ relative to the paramagnetic state.

The above related discussion has been added to the revised version.

- 3. The surface projected dispersions of the paramagnetic state along cut1 are shown in Fig. 4(d), which has been compared with experimental results in Fig. 4(e-f), but I am confused with the magnetic state of the measured constant energy surfaces and dispersions, and the authors should give the results of the surface projected dispersions of ferrimagnetic state.*

I agree with reviewer #1 that the calculations of the surface projected dispersions of ferrimagnetic state will help us better understand the measured constant energy surfaces and dispersions, but the associated DFT calculations are huge. This is because ferrimagnetism requires a larger number of cells and slab layers to be considered compared to paramagnetic state. In this case, both the amount of computation and the memory required have exceeded the limits of current supercomputing. In addition, more importantly, we believe that the absence of relevant DFT calculation results will not affect the overall conclusion of our work.

Reviewer #2 Comments:

The manuscript by Li et.al. presents a comprehensive study of the electronic band structure of NdAlSi, a compound which has been recently discussed as a magnetic Weyl semimetal candidate. The study aims to unveil a possible regulation of Weyl fermions by the emergence of a magnetic ground state below a critical temperature. While the authors do an excellent job in mapping the electronic band structure and surface states of their in-situ cleaved NdAlSi crystals -strongly suggesting the existence of a Weyl Semimetal phase by carrying out well-known ARPES protocols-, the evidence for a “regulation” of Weyl fermions through a ferrimagnetic ground state is not clear from their experiments. From the electronic band calculations, when the crystal is in the ferrimagnetic state, there is an effect of band folding and consequently the natural creation of additional Weyl points visible in the folded BZ. In addition, the authors observe an increase of the spectral weight in one of the energy cuts while going through the magnetic transition temperature, which they attribute to the magnetic interactions in the crystal. However, these findings are too incipient to argue for a “regulation of Weyl Fermions” by magnetic ground states. The connection between magnetism and Weyl physics has to be further elaborated to support the claims. Therefore, the authors should either look for more convincing evidence of an active interplay between magnetism and Weyl fermions, or reformulate/revisit their main claims putting more emphasis on the band structure study, where they did a good job. I cannot recommend publication of the article in the present form, but I think there is room for improvement. In particular, through revision of the following major issues:

1) The first evidence of bandstructure changes by magnetism is provided by calculations of the UUD ferrimagnetic state of NdAlSi. Due to the band-folding effect, it is not straightforward to judge whether there is an emergence of new Weyl points, or just a redistribution of the WPs at different k space locations caused by band folding. The authors should mark in the constant energy contours in Fig. 1 (g-l) which new WPs are expected to appear as a result of magnetic ordering, besides the ones projected on the folded BZ. In the manuscript there is always a notion of “generation of new Weyl points”, which should not be confused by projecting the existing WPs on other k locations in the BZ.

2) The physics underlying the effect of magnetic ordering in the bandstructure of NdAlSi. In magnetic solids, the most notable effect from a magnetic ground state is the spin-splitting of electronic bands, and in some cases, a shift of the degeneracy points in momentum space can be

inferred (e.g. a recent paper by Piva et. al. on another WSM candidate CeAlSi, 10.1103/PhysRevResearch.5.013068). The authors should describe in detail what are the crucial factors that are responsible for a change in the bandstructure (i.e. band folding) in the ferrimagnetic phase: i) the mere fact that you have exchange interactions between the Nd atoms; ii) a potential change in atomic bond lengths, iii) any type of surface reconstruction that may happen at low temperatures. A surface reconstruction has been shown (see. e.g. Advanced Materials 33 (21), 2008634, 2021) to induce band folding and create “new” Weyl features on the folded bandstructure). The authors should refer to this publication and discuss other origins of band folding other than magnetic interactions.

3) In conjunction with the first two points, a figure comparing the experimental k_x - k_y contour (e.g. Fig. 4b) at the same binding energy and photon energy of the paramagnetic and ferrimagnetic UUD phase should be constructed together with the calculated contours at the same energies, to see more clearly the fingerprints of band folding in the spectra (not easy to compare the symmetry of the BZ in the experimental contours).

4) In several passages of the manuscript it is mentioned that new Weyl points are created in the UUD phase, but in Fig. 4c the additional weak feature cannot be safely attributed to be a Weyl point. The authors should make an effort to corroborate this important information, either by refining the measurements (Fig. 4h is just a blurry contrast with no structure to match with the green line) or by looking at other relevant energies/momenta where this effect also take place.

5) In the present form, the title is misleading and oversells the content of the manuscript. Even in the best case scenario, where band folding and new Weyl points can be unambiguously ascribed to the ferrimagnetic phase, the “regulation of Weyl Fermions” is still far fetched. “Emergence of Weyl points in the ferrimagnetic phase of NdAlSi” or something along those lines will be more appropriate.

Minor issues:

1) Introduction: For a more general reading audience, it should be briefly commented why breaking of both TRS and IS still holds a Weyl Semimetal phase. Usually it is one or the other, not both at the same time.

2) The authors present a “low quality surface” measurement to suppress the surface states and just be sensitive to the bulk electronic dispersion. It would be instructive for the surface science

community to describe how they prepare this low quality surface, whether it is a cleavage effect or rather a post-cleavage surface treatment in vacuum. Alternatively one can distinguish bulk and surface states by performing photon energy dependent measurements. I would be curious if the bulk states (and WP dispersions) can be also corroborated that way.

3) Figure 1 (calculations) and Figure 4 (experiments) contain important information about the electronic structure of the ferrimagnetic state, which is the novelty of the manuscript. It is difficult for the reader to jump back and forth to follow the content, so the idea of first discussing the PM phase and then only the effect of the FM is more convenient for the manuscript outline (and one avoids repetitive content at the discussions of Fig. 1 and 4). Especially for major issue 3) putting the calculated k_x - k_y next to the spectra of Figs 4b and 4c will help a lot to identify the relevance of band folding and the new electronic features.

Response to Reviewer #2

The manuscript by Li et.al. presents a comprehensive study of the electronic band structure of NdAlSi, a compound which has been recently discussed as a magnetic Weyl semimetal candidate. The study aims to unveil a possible regulation of Weyl fermions by the emergence of a magnetic ground state below a critical temperature. While the authors do an excellent job in mapping the electronic band structure and surface states of their in-situ cleaved NdAlSi crystals -strongly suggesting the existence of a Weyl Semimetal phase by carrying out well-known ARPES protocols-, the evidence for a “regulation” of Weyl fermions through a ferrimagnetic ground state is not clear from their experiments. From the electronic band calculations, when the crystal is in the ferrimagnetic state, there is an effect of band folding and consequently the natural creation of additional Weyl points visible in the folded BZ. In addition, the authors observe an increase of the spectral weight in one of the energy cuts while going through the magnetic transition temperature, which they attribute to the magnetic interactions in the crystal. However, these findings are too incipient to argue for a “regulation of Weyl Fermions” by magnetic ground states. The connection between magnetism and Weyl physics has to be further elaborated to support the claims. Therefore, the authors should either look for more convincing evidence of an active interplay between magnetism and Weyl fermions, or reformulate/revisit their main claims putting more emphasis on the band structure study, where they did a good job. I cannot recommend publication of the article

in the present form, but I think there is room for improvement. In particular, through revision of the following major issues:

We thank Reviewer #2 for the careful review and constructive comments. It is encouraging that Reviewer #2 has captured the main results and the significance of our work and sees room for improvement. We have followed Reviewer #2's suggestions and made substantial modifications to the revised version of the manuscript.

In the following, we give our response to the Reviewer's comments one by one.

major issues:

- 1. The first evidence of bandstructure changes by magnetism is provided by calculations of the UUD ferrimagnetic state of NdAlSi. Due to the band-folding effect, it is not straightforward to judge whether there is an emergence of new Weyl points, or just a redistribution of the WPs at different k space locations caused by band folding. The authors should mark in the constant energy contours in Fig. 1 (g-l) which new WPs are expected to appear as a result of magnetic ordering, besides the ones projected on the folded BZ. In the manuscript there is always a notion of "generation of new Weyl points", which should not be confused by projecting the existing WPs on other k locations in the BZ.*

In NdAlSi, the effect of ferrimagnetism on the Weyl fermions is mainly reflected in two parts. The first part is that the net magnetic moment causes the time reversal symmetry (TRS) breaking and drives a shift of the Weyl points in the 3D BZ, which is also expected to be present in CeAlSi. The second part is that the ferrimagnetic order leads to the folding of the BZ and generates new Weyl points in the BZ relative to the paramagnetic state. For a better understanding of these points, we have made some changes to Figure 1, as shown in Fig. R2_1. It is worth noting that the new Weyl points just resulting from BZ folding by ferrimagnetism which are represented by hollow points in Fig. 1i-1l. As for whether the Weyl points induced by BZ folding can be called new Weyl points? I would however strongly argue that a Weyl point that emerges at a momentum location where there was previously no Weyl point is a new Weyl point.

Fig. R2_1. The revised figure of Figure 1.

2. *The physics underlying the effect of magnetic ordering in the bandstructure of NdAlSi. In magnetic solids, the most notable effect from a magnetic ground state is the spin-splitting of electronic bands, and in some cases, a shift of the degeneracy points in momentum space can be inferred (e.g. a recent paper by Piva.et. al. on another WSM candidate CeAlSi, 10.1103/PhysRevResearch.5.013068). The authors should describe in detail what are the crucial factors that are responsible for a change in the bandstructure (i.e. band folding) in the ferrimagnetic phase: i) the mere fact that you have exchange interactions between the Nd atoms; ii) a potential change in atomic bond lengths, iii) any type of surface reconstruction that may happen at low temperatures. A surface reconstruction has been shown (see. e.g. Advanced Materials 33 (21), 2008634, 2021) to induce band folding and create “new” Weyl features on the folded bandstructure). The authors should refer to this publication and discuss other origins of band folding other than magnetic interactions.*

As mentioned above, in NdAlSi, the effect of ferrimagnetism on the Weyl fermions is mainly reflected in two parts. The first part is that the net magnetic moment causes the time reversal symmetry (TRS) breaking and drives the Weyl point shift in the 3D BZ. The second part is the

generation of new Weyl points caused by the folding of the BZ in the ferrimagnetic state as compared to the paramagnetic state. In fact, ferrimagnetism in NdAlSi comes from the f orbital of Nd, and the interaction between Weyl fermions and magnetism is mainly of Ruderman-Kittel-Kasuya-Yosida (RKKY) type, the magnitude of which oscillates with the distance (R) between the Nd atoms (New J. Phys. 25 (2023) 013037).

We agree with Reviewer #2 that there are other possible origins of band folding, such as surface reconstruction. The paper (Advanced Materials 33 (21), 2008634, 2021) mentioned by the Reviewer #2 gives a very good example of surface reconstruction achieved by preparing a dangling-bond-free on P-terminated surface which suppresses the trivial surface states of NbP. However, in NdAlSi, no structural phase transitions have been reported except for magnetic phase transitions in the temperature range from 2 K to 15 K. The probability of spontaneous surface reconstruction occurring in the temperature range from 2 K to 15 K and forming a triple charge density wave (CDW) exactly along the in-plane diagonal direction should be very small. But this band folding behavior can be explained naturally by magnetic interactions. Therefore, we find the magnetic origin of band folding more plausible.

Changes related to the above discussion have been added to the revised version.

- 3. In conjunction with the first two points, a figure comparing the experimental k_x - k_y contour (e.g. Fig. 4b) at the same binding energy and photon energy of the paramagnetic and ferrimagnetic UUD phase should be constructed together with the calculated contours at the same energies, to see more clearly the fingerprints of band folding in the spectra (not easy to compare the symmetry of the BZ in the experimental contours).*

I agree with reviewer #2 that the figure comparing the experimental k_x - k_y contour together with the calculated contours at the same energies will help us better understand the measured constant energy surfaces. The constant energy surfaces calculation in the paramagnetic state has been added to Fig. 4, but not for ferrimagnetic UUD phase. This is because the associated DFT calculations in ferrimagnetic state are huge which requires a larger number of cells and slab layers to be considered compared to paramagnetic state. In this case, both the amount of computation and the

memory required have exceeded the limits of current supercomputing. Therefore, I only added the calculation results in the paramagnetic state for comparison, as shown in Fig. R2_2.

- 4. In several passages of the manuscript it is mentioned that new Weyl points are created in the UUD phase, but in Fig. 4c the additional weak feature cannot be safely attributed to be a Weyl point. The authors should make an effort to corroborate this important information, either by refining the measurements (Fig. 4h is just a blurry contrast with no structure to match with the green line) or by looking at other relevant energies/momenta where this effect also take place.*

In Fig. 4c, the additional weak feature ($E_b=70$ meV, it is not the energy level of the Weyl points) should be the constant energy contour structure caused by the UUD ferrimagnetic order. We use the appearance of this feature to prove that there is regulation of ferrimagnetic order in NdAlSi, but not try to attribute it to be a Weyl point. To make the features in Figure 4h more obvious, we removed the green lines in the revised version, as shown in Fig. R2_2.

It is worth noting that this measurement is very challenging, on the one hand, because of the strong three-dimensional nature of single crystals, which is not easy to cleavage. On the other hand, its ferromagnetic transition temperature is very low (only 3.5 K), and very few ARPES systems in the world reach this temperature. The current data is the best quality available.

Fig. R2_2. The revised figure of Figure 4.

5. *In the present form, the title is misleading and oversells the content of the manuscript. Even in the best case scenario, where band folding and new Weyl points can be unambiguously ascribed to the ferrimagnetic phase, the “regulation of Weyl Fermions” is still far fetched. “Emergence of Weyl points in the ferrimagnetic phase of NdAlSi” or something along those lines will be more appropriate.*

While we believe that the two effects on the Weyl fermions caused by ferrimagnetism is covered by our previous title, we suggest a more restricted title that we hope is more in line with the reviewer’s view: “Emergence of Weyl Fermions by Ferrimagnetism in a Noncentrosymmetric Magnetic Weyl Semimetal”.

Minor issues

- 1. Introduction: For a more general reading audience, it should be briefly commented why breaking of both TRS and IS still holds a Weyl Semimetal phase. Usually it is one or the other, not both at the same time.*

We have added some discussion of this fact to the introduction.

- 2. The authors present a “low quality surface” measurement to suppress the surface states and just be sensitive to the bulk electronic dispersion. It would be instructive for the surface science community to describe how they prepare this low quality surface, whether it is a cleavage effect or rather a post-cleavage surface treatment in vacuum. Alternatively one can distinguish bulk and surface states by performing photon energy dependent measurements. I would be curious if the bulk states (and WP dispersions) can be also corroborated that way.*

The low-quality surface is a cleavage effect. NdAlSi is a three-dimensional non-centrosymmetric crystal, the single crystal after cleavage shows flat areas and areas with multiple steps. The corresponding location can be found by spatial scanning of the sample. According to our experience, it is worth noting that relative to the Nd atom terminated surface of the Nd-Al cleavage plane, the corresponding Al atom terminated surface of the Nd-Al cleavage plane is less likely to show large flat areas.

In general, photon energy dependent measurements are often used to distinguish between bulk state and surface states. We typically also employ this to distinguish between surface states and bulk states, where the electronic structure of bulk state is dispersive along the K_z direction, while the surface state is not. However, it is currently a challenging work for NdAlSi, because NdAlSi has a complex surface state electronic structure and the surface state signal in NdAlSi is very strong, while the bulk state has a smaller proportion in BZ (Fig. R2_3a-3b) and overlap with the surface states which makes it difficult to distinguish the bulk state from the surface state. We have also tried to do the photon energy dependent measurements as shown in Fig. R2_3c. However, due to the limitation of measurement accuracy, it is difficult to give a convincing conclusion on the current data, so we do not discuss it in the manuscript.

Fig. R2_3. (a-b) Calculated bulk Fermi surface at the $k_z \sim 0$ (a) and 0.67 (b) π/c plane. (c) Photon-energy-dependent measurements of NdAlSi crossing the zone center along the cut direction (red line) in (d) at Fermi level. (d) The Fermi surface measure with photon energy of 30 eV, the red line defined the cut direction of the photon-energy-dependent measurements.

3. *Figure 1 (calculations) and Figure 4 (experiments) contain important information about the electronic structure of the ferrimagnetic state, which is the novelty of the manuscript. It is difficult for the reader to jump back and forth to follow the content, so the idea of first discussing the PM phase and then only the effect of the FM is more convenient for the*

manuscript outline (and one avoids repetitive content at the discussions of Fig. 1 and 4). Especially for major issue 3) putting the calculated k_x - k_y next to the spectra of Figs 4b and 4c will help a lot to identify the relevance of band folding and the new electronic features.

For the convenience of the reader and in line with the reviewer's suggestion, the distributions of Weyl fermions from the top view of the BZ in the ferrimagnetic state are replotted in Fig. 4a-4d. The updated version is shown in Fig. R2_2 above.

Reviewer #3 Comments:

Referee Report for NCOMMS-23-17139-T, “Ferrimagnetic Regulation of Weyl Fermions in a Noncentrosymmetric Magnetic Weyl Semimetal” by C. Li et al., is being considered for publication in Nature Communication.

This ARPES work is a tour de force with parts of the data taken at 2K. The manuscript aims to address the interplay between magnetism and topological orders in the rare earth RAlX (R: Rare earth; X: Si or Ge) family, with a specific focus on NdAlSi as a magnetic Weyl semimetal candidate, and the existence of Weyl fermions regulated by ferrimagnetism. The paper provides a systematic investigation into the electronic structure and topological properties of NdAlSi using angle-resolved photoemission spectroscopy (ARPES) measurements and density functional theory (DFT) calculations, with implications for the development of new materials for spintronics applications. The paper was written in a very pedagogical and concise manner, making it an enjoyable read. However, it was not clear from the text if ferrimagnetism plays any role different from ferromagnetism in Weyl semimetals. The authors pointed out in page 4 that the UUD ferrimagnetic phase would reconstruct the paramagnetic BZ, leading to folding of bands and creation of new Weyl nodes. I suggest the authors emphasize its contrast to ferromagnetic Weyl semimetals, such as PrAlGe and CeAlGe, where the onset of magnetism would only shift the Weyl nodes in k-space and split the bands due to Zeeman effect.

One of the biggest challenges for measuring magnetic materials with ARPES is finding a single domain. Typically, the magnetic domains are micron-size and therefore smaller than the beam spot size in most ARPES setups. Unless being field cooled, the sample is likely in a mixed state with UUD and DDU domains at 2K. However, Fig.4 seems to imply that the ARPES measurements at 2K were taken from a dominating magnetic domain. Could the authors please clarify this in the manuscript?

Figure 4 is quite confusing in comparing the dispersions measured at 15K vs 2K. The DFT overlay in Fig.4e is clearly not matching the data. The bands overlaid by the green lines in Fig.4f are also present in Fig.4e, but they were not captured by the DFT calculation. Comparing with surface

projected DFT in Fig.4d, it seems to me that this “green-line band” is a surface state. However, looking at the raw data, I would argue that Fig.4e and Fig.4f are almost identical. It seems to me that the band overlaid by the green lines could also be equally well explained by a surface state. Even in the zoomed in figures, I find it hard to difficult to see the second set of bands under the green dashed lines. Could the authors please clarify? Perhaps second derivatives to the raw data, or a change of color scale would help?

The center electron pocket in Fig.4 has “filled up” intensity in the FS plot, instead of a closed loop. This is typically due to blurring from the k_z dispersion. However, it is clear from the surface projected DFT calculation that this electron pocket is a surface state. I also see that the other bands are reasonably sharp, and therefore the blurring cannot come from lack of surface quality. This is even clearer in Fig.S9, where the center electron pocket is much broader than the other bands. Do the authors understand why the blurring of intensity only happens to this electron pocket?

In summary, I find the current work highly significant and well written. However, I have my reservations about the ARPES features the authors ascribed to the ferrimagnetic onset. In particular, the generation of new Weyl points in the ferrimagnetic phase is not clear to me. I would recommend another round of discussion before making a decision on publication.

Response to Reviewer #3

This ARPES work is a tour de force with parts of the data taken at 2K. The manuscript aims to address the interplay between magnetism and topological orders in the rare earth RAlX (R: Rare earth; X: Si or Ge) family, with a specific focus on NdAlSi as a magnetic Weyl semimetal candidate, and the existence of Weyl fermions regulated by ferrimagnetism. The paper provides a systematic investigation into the electronic structure and topological properties of NdAlSi using angle-resolved photoemission spectroscopy (ARPES) measurements and density functional theory (DFT) calculations, with implications for the development of new materials for spintronics applications. The paper was written in a very pedagogical and concise manner, making it an enjoyable read. However, it was not clear from the text if ferrimagnetism plays any role different from ferromagnetism in Weyl semimentals. The authors pointed out in page 4 that the UUD ferrimagnetic phase would reconstruct the paramagnetic BZ, leading to folding of bands and

creation of new Weyl nodes. I suggest the authors emphasize its contrast to ferromagnetic Weyl semimetals, such as PrAlGe and CeAlGe, where the onset of magnetism would only shift the Weyl nodes in k -space and split the bands due to Zeeman effect.

We thank Reviewer #3 for reviewing our manuscript and for the encouraging feedback.

In NdAlSi, the effect of ferrimagnetism on Weyl fermions is mainly reflected in two parts. The first part is that the net magnetic moment causes the time reversal symmetry (TRS) breaking and drives a shift of the Weyl points in the 3D BZ, which is also expected to be present in CeAlSi. The second part is that the ferrimagnetic order leads to a folding of the BZ and generates new Weyl points in the BZ relative to the paramagnetic state.

A related description has been added to the revised version.

One of the biggest challenges for measuring magnetic materials with ARPES is finding a single domain. Typically, the magnetic domains are micron-size and therefore smaller than the beam spot size in most ARPES setups. Unless being field cooled, the sample is likely in a mixed state with UUD and DDU domains at 2K. However, Fig.4 seems to imply that the ARPES measurements at 2K were taken from a dominating magnetic domain. Could the authors please clarify this in the manuscript?

We thank Reviewer #3 for pointing out the magnetic domain problem. We have also considered the existence of two different magnetic domains in NdAlSi, but since the results of the DFT calculation of the two domains are very close, they cannot be distinguished based on the current ARPES resolution, as shown in Fig. R3_1. For simplicity, we only take the UUD type magnetic domains as an example. The relevant instructions have been added to the revised version.

Fig. R3_1. The DFT calculations of UUD and DDU type magnetic domains along C- Γ -Z direction.

Figure 4 is quite confusing in comparing the dispersions measured at 15K vs 2K. The DFT overlay in Fig.4e is clearly not matching the data. The bands overlaid by the green lines in Fig.4f are also present in Fig.4e, but they were not captured by the DFT calculation. Comparing with surface projected DFT in Fig.4d, it seems to me that this “green-line band” is a surface state. However, looking at the raw data, I would argue that Fig.4e and Fig.4f are almost identical. It seems to me that the band overlaid by the green lines could also be equally well explained by a surface state. Even in the zoomed in figures, I find it hard to difficult to see the second set of bands under the green dashed lines. Could the authors please clarify? Perhaps second derivatives to the raw data, or a change of color scale would help?

We also thank Reviewer #3 for his constructive comments and suggestions to improve our paper. I fully agree that the DFT calculation does not match the data exactly quantitatively but can explain the characteristics of the band qualitatively. In order not to create confusion about overlapping, we adjust them to left-right contrast. We also noticed that the bulk state of the ferrimagnetic state overlaps significantly with the surface state of paramagnetic state. Therefore, the most promising

reflection of the effect of the ferrimagnetic order should be the characteristic changes in the pink region. To clarify it, we have updated Fig. 4, as shown in Fig. R3_2.

Fig. R3_2. The revised figure of Figure 4.

The center electron pocket in Fig.4 has “filled up” intensity in the FS plot, instead of a closed loop. This is typically due to blurring from the k_z dispersion. However, it is clear from the surface projected DFT calculation that this electron pocket is a surface state. I also see that the other bands are reasonably sharp, and therefore the blurring cannot come from lack of surface quality. This is even clearer in Fig.S9, where the center electron pocket is much broader than the other bands. Do the authors understand why the blurring of intensity only happens to this electron pocket?

According to the DFT calculations (Fig. R3_3d), the center electron pocket in Fig. 4 is the surface band mainly from Nd atom of the Nd terminated surface of the Nd-Al cleavage plane. During the measurement, some gas molecules absorbed on the sample surface over time, causing the sample to age and thus blurring the center electron pocket. In fact, in addition to the center electron pocket, the electron pockets (also from Nd atom of the Nd terminated surface of the Nd-Al cleavage plane) along the X-M direction are also blurred over time (Fig. R3_3a-3c).

Fig. R3_3. (a-c) The constant energy contours at binding energy of 70 meV measured with the photon energy of 41 eV at 15 K on an as cleaved surface (a), 2 K (b) and 15 K (c). The constant energy contours are measured at different times after the sample is cleaved. (d) Calculated band structure of NdAlSi along high-symmetry directions and their atomic nature.

Summary of changes:

1. Based on the comment from Reviewer #1 and Reviewer #2 as well as Reviewer #3, on Page 7 in the manuscript, we have added a description about ferrimagnetic regulation of Weyl fermions: *The ferrimagnetism in NdAlSi comes from the f orbital of Nd atoms, and the interaction between Weyl fermions and magnetism is mainly of Ruderman-Kittel-Kasuya-Yosida (RKKY) type, the magnitude of which oscillates with the distance between the Nd atoms[68]. The results from the DFT bulk calculations show that the effect of ferrimagnetism on Weyl fermion is mainly reflected in two parts. The first part is that the net magnetic moment causes the TRS breaking and drives a shift of the Weyl points in the 3D BZ, which is also expected to be observed in CeAlSi[43, 46]. However, the actual shift of the Weyl point in 3D BZ is not easy to be measured according to the current accuracy. The second part is that the ferrimagnetic order leads to the folding of the BZ and generates new Weyl points in the BZ relative to the paramagnetic state.*
2. Based on the comment from Reviewer #2, on Page 17 in the manuscript, we have updated Fig.
3. Based on the comment of Reviewer #2, on page 8 in the manuscript, we have added a discussion of the possibility that the new features are caused by surface reconfiguration: *This feature is not present at 15 K (Fig. 4f) as well as the surface projected DFT calculations in paramagnetic state (Fig. 4e). First, we conclude that the new feature is unlikely to be caused by surface reconstruction, although surface reconstruction has previously been reported to produce or suppress some Fermi surface features[69]. Since no structural phase transitions have been reported except for magnetic phase transitions in the temperature range from 2 K to 15 K in NdAlSi. The probability of spontaneous surface reconstruction occurring in the temperature range from 2 K to 15 K and forming a triple charge density wave (CDW) exactly along the in-plane diagonal direction should be very small.*
4. Based on the comment of Reviewer #2, We have adjusted the title to: *“Emergence of Weyl Fermions by Ferrimagnetism in a Noncentrosymmetric Magnetic Weyl Semimetal”*.
5. Based on the comment of Reviewer #2, on Page 2-3 in the manuscript, we have added some related discussion in the introduction: *In general, the establishment of Weyl semimetal must break either inversion or time reversal symmetry. (Page 2) Above the magnetic transition temperature, the RAlX family are already an IS broken Weyl semimetal. When the temperature*

is lowered below the magnetic transition temperature, the magnetic structure will affect the existing Weyl fermions as well as generate additional Weyl fermions. Therefore, the RAlX family provides an appropriate platform to study the interaction between magnetism and Weyl fermions. (Page 3)

6. Based on the comment of Reviewer #2 and Reviewer #3, on Page 20 in the manuscript, we have updated Fig. 4.
7. Based on the comment of Reviewer #2, on Page 5 in the Supplementary Material, we have added a description of the acquisition of low-quality surface: *It is worth noting that low quality surface acquisition is a cleavage effect. NdAlSi is a three-dimensional non-centrosymmetric crystal, the single crystal after cleavage has a certain probability of showing a flat surface, a bad surface, and a surface in between. The corresponding location can be found by spatial scanning of the sample.*
8. Based on the comment of Reviewer #3, on Page 2-3 in the Supplementary Material, we have added a description of magnetic domain in NdAlSi.
9. On Page 18 in the manuscript, we corrected the orientation of the calculated Fermi surface in Fig. 2c (It's rotated 90 degrees).
10. On Page 11 in the Supplementary Material, we corrected the correspondence between the band calculations and the termination surface (Fig. S4e and S4f) (The previous one was reversed).
11. On Page 13 in the Supplementary Material, we corrected the orientation of the calculated Fermi surface in Fig. S6a-b (It's rotated 90 degrees).
12. All changes are marked in red in the revised version.

REVIEWERS' COMMENTS

Reviewer #1 (Remarks to the Author):

The authors have properly addressed all my questions and I would suggest its publication in Nature Communication.

Reviewer #2 (Remarks to the Author):

The authors have addressed the vast majority of my inquiries and have revised the manuscript accordingly, which has led to a substantial improvement of the presented work. In particular, they have acknowledged my suggestion and changed the title to avoid overselling, putting it in a more realistic connection to the obtained results. There are still some minor issues that are not fully addressed but I would rate the manuscript as publishable in the present form.

Reviewer #3 (Remarks to the Author):

Referee Report for NCOMMS-23-17139-T Revision 1, "Emergence of Weyl Fermions by Ferrimagnetism in a Noncentrosymmetric Magnetic Weyl Semimetal" by C. Li et al., is being considered for publication in Nature Communication.

The Authors have carefully replied to all of the questions I raised in the report. In my opinion, there are several results in this study that could interest the broad readership of Nature Comms.

1. The RAISi system is notoriously known to suffer from site disorders, where the Al and Si atoms randomly occupy the same Wyckoff positions leading to a centrosymmetric crystallographic group. It has been under debate whether the Weyl nodes would still exist in these compounds or not [Phys. Rev. B 102, 085143 (2020)]. The fact that the Authors observed topological surface Fermi arcs in the paramagnetic phase is interesting and valuable in itself.

2. It is very challenging to handle 4f electrons in magnetic materials in DFT, let alone calculating slabs with rare-earth element termination. Therefore, it is important to have high quality ARPES data to guide future experimental studies. This work has done a thorough investigation in both paramagnetic and magnetic phases. Considering this family of compounds have attracted a lot of attention recently due to the Weyl semimetal physics, this work could be useful for future references.

3. Searching for ARPES signatures across magnetic phase transition is typically a difficult task, partly due to the formation of magnetic domains that are smaller than the typical beam spot size. In relation to the magnetic Weyl semimetal, many attempts have been made to study the shift of Weyl nodes across the magnetic transition. However, this is intrinsically difficult due to the smallness of the Zeeman energy. The Authors have come up with a novel attempt, that is to look for evidence of BZ folding in a ferrimagnet. Having said that, the experimental evidence of the BZ folding is weak in my opinion. I could not convince myself that there are additional bands in Fig.4l and Fig.S11e. I have reached a similar conclusion from the MDC cuts, the peak positions marked in green circles in Fig.S11h are not so different from the noise in Fig.11g and 11i. On the other hand, I agree with the Authors that the fresh cleaved and thermal cycled 15K data in Fig.S11d and S11f share common features, while the 2K data seem more "blurred" near the Fermi energy. This would be more consistent with the band folding scenario.

In conclusion, I find the current manuscript shares novel ideas and high-quality data that would be interesting to a broad audience. I would recommend the publication of this manuscript in Nature Comms.

Response to Reviewer's Comments

Reviewer #1 Comments:

The authors have properly addressed all my questions and I would suggest its publication in Nature Communication.

Response to Reviewer #1

We also thank him/her for recommending our paper for publication in Nature Communications.

Reviewer #2 Comments:

The authors have addressed the vast majority of my inquiries and have revised the manuscript accordingly, which has led to a substantial improvement of the presented work. In particular, they have acknowledged my suggestion and changed the title to avoid overselling, putting it in a more realistic connection to the obtained results. There are still some minor issues that are not fully addressed but I would rate the manuscript as publishable in the present form.

Response to Reviewer #2

We thank Reviewer #2 for reviewing our paper again and thank him/her for recommending our paper for publication in Nature Communications.

Reviewer #3 Comments:

Referee Report for NCOMMS-23-17139-T Revision 1, "Emergence of Weyl Fermions by Ferrimagnetism in a Noncentrosymmetric Magnetic Weyl Semimetal" by C. Li et al., is being considered for publication in Nature Communication.

The Authors have carefully replied to all of the questions I raised in the report. In my opinion, there are several results in this study that could interest the broad readership of Nature Comms. 1. The RAISi system is notoriously known to suffer from site disorders, where the Al and Si atoms randomly occupy the same Wyckoff positions leading to a centrosymmetric crystallographic group. It has been under debate whether the Weyl nodes would still exist in these compounds or not [Phys. Rev. B 102, 085143 (2020)]. The fact that the Authors observed topological surface Fermi arcs in the paramagnetic phase is interesting and valuable in itself.

2. It is very challenging to handle 4f electrons in magnetic materials in DFT, let alone calculating slabs with rare-earth element termination. Therefore, it is important to have high quality ARPES data to guide future experimental studies. This work has done a thorough investigation in both paramagnetic and magnetic phases. Considering this family of compounds have attracted a lot of attention recently due to the Weyl semimetal physics, this work could be useful for future references.

3. Searching for ARPES signatures across magnetic phase transition is typically a difficult task, partly due to the formation of magnetic domains that are smaller than the typical beam spot size. In relation to the magnetic Weyl semimetal, many attempts have been made to study the shift of Weyl nodes across the magnetic transition. However, this is intrinsically difficult due to the smallness of the Zeeman energy. The Authors have come up with a novel attempt, that is to look for evidence of BZ folding in a ferrimagnet. Having said that, the experimental evidence of the BZ folding is weak in my opinion. I could not convince myself that there are additional bands in Fig.4l and Fig.S11e. I have reached a similar conclusion from the MDC cuts, the peak positions marked in green circles in Fig.S11h are not so different from the noise in Fig.11g and 11i. On the other hand, I agree with the Authors that the fresh cleaved and thermal cycled 15K data in Fig.S11d and S11f share common features, while the 2K data seem more “blurred” near the Fermi energy. This would be more consistent with the band folding scenario.

In conclusion, I find the current manuscript shares novel ideas and high-quality data that would be interesting to a broad audience. I would recommend the publication of this manuscript in *Nature Comms*.

Response to Reviewer #3

We thank Reviewer #3 for reviewing our paper again, and for capturing the challenge and significance of our work. We also thank him/her for recommending our paper for publication in *Nature Communications*.